# Treatment of Overlap Syndromes in Autoimmune Liver Disease: A Systematic Review and Meta-Analysis

**DOI:** 10.3390/jcm9051449

**Published:** 2020-05-13

**Authors:** Benjamin L. Freedman, Christopher J. Danford, Vilas Patwardhan, Alan Bonder

**Affiliations:** 1Department of Medicine, Beth Israel Deaconess Medical Center, 330 Brookline Ave., Boston, MA 02215, USA; bfreedm1@bidmc.harvard.edu; 2Division of Gastroenterology and Hepatology, Beth Israel Deaconess Medical Center, 330 Brookline Ave., Dana 603, Boston, MA 02215, USA; cdanford@bidmc.harvard.edu; 3Liver Center, Autoimmune and Cholestatic Liver Disease Program, Department of Medicine, Beth Israel Deaconess Medical Center, Harvard Medical School, 110 Francis St. Suite 8E, Boston, MA 02215, USA; vpatward@bidmc.harvard.edu

**Keywords:** overlap syndrome, autoimmune liver disease, immunosuppression, corticosteroid, ursodeoxycholic acid

## Abstract

The treatment of overlap syndromes is guided by small observational studies whose data have never been synthesized in a rigorous, quantitative manner. We conducted a systematic review and meta-analysis to evaluate the efficacy of available treatments for these rare and morbid conditions. We searched the literature for studies comparing ≥2 therapies for autoimmune hepatitis (AIH)-primary biliary cholangitis (PBC), AIH-primary sclerosing cholangitis (PSC), PBC-PSC, AIH-PBC-PSC, autoimmune cholangitis (AIC), or autoimmune sclerosing cholangitis (ASC) with respect to various clinical outcomes, including biochemical improvement and transplant-free survival. A total of 28 studies met the inclusion criteria for AIH-PBC, AIH-PSC, AIC, and ASC. AIH-PBC patients tended to experience more biochemical improvement with ursodeoxycholic acid (UDCA) + [corticosteroids and/or antimetabolites], i.e., “combination therapy”, than with corticosteroids ± azathioprine (RR = 4.00, 95% CI 0.93–17.18). AIH-PBC patients had higher transplant-free survival with combination therapy than with UDCA, but only when studies with follow-up periods ≤90 months were excluded (RR = 6.50, 95% CI 1.47–28.83). Combination therapy may therefore be superior to both UDCA and corticosteroids ± azathioprine for the treatment of AIH-PBC, but additional studies are needed to show this definitively and to elucidate optimal treatments for other overlap syndromes.

## 1. Introduction

The term “overlap syndromes” is used to describe a family of rare, morbid conditions characterized by biochemical, immunologic, histologic, or cholangiographic features of more than one of the well-recognized autoimmune liver diseases: Autoimmune hepatitis (AIH), primary biliary cholangitis (PBC), and primary sclerosing cholangitis (PSC) [1]. These overlap syndromes include AIH-PBC, AIH-PSC, PBC-PSC, AIH-PBC-PSC, autoimmune cholangitis (AIC), and autoimmune sclerosing cholangitis (ASC). While their initial presentations differ, any of them may ultimately progress to cirrhosis and its sequelae.

AIH-PBC is the most studied of these disorders, occurring in 1–3% of patients with PBC [2] and 7% of patients with AIH [3]. Based upon histologic, biochemical, and/or immunologic features of each parent disorder, the Paris criteria are the most widely accepted means of diagnosing AIH-PBC [4]. Most investigators to date have chosen a first-line treatment regimen including both ursodeoxycholic acid (UDCA) and corticosteroids, with or without azathioprine (AZA), often with encouraging results [2,4,5,6,7,8,9,10,11,12]. However, in some studies patients appeared to benefit from UDCA alone [7,10,13], or from just corticosteroids with or without AZA [3,14]. UDCA has been augmented with tacrolimus [10], mycophenolate mofetil (MMF) [15,16], or cyclosporine [8,10,17] as second line therapies, with some success, in patients intolerant of or unresponsive to corticosteroids and/or AZA. At 10 years, progression to cirrhosis among patients with AIH-PBC is approximately 44–48% [18,19], and transplant-free survival is 52–92% [19,20,21].

AIC, also known as AIH-cholestatic overlap syndrome, is typically defined by the coexistence of AIH and a cholestatic syndrome that does not meet criteria for either classic PBC or PSC: alkaline phosphatase (AP) and γ-glutamyltransferase (GGT) are elevated and liver biopsy shows bile duct injury (in various histologic patterns), but anti-mitochondrial antibodies (AMA) are absent and cholangiography is normal [1]. Czaja et al. therefore postulate that, in some cases of AIC, AMA-negative PBC or small-duct PSC may be the true clinical entities overlapping with AIH [1]. AIC has been identified in 1% and 11% of patients with PBC [22] and AIH [3], respectively. The same first line treatment options exist for AIC as for AIH-PBC (UDCA [23], corticosteroids ± AZA [3,22,24], or a combination of both [23,25]), although AIC has shown a poorer response to treatment across the board. Limited prognostic data suggest poor clinical outcomes as in one study, two out of six patients with AIC experienced either liver-related death or transplant after a median of three months [3] and in another, one of 20 patients with AIC died from liver failure seven months into follow-up [24].

The diagnosis of AIH-PSC requires the coexistence in adult patients of AIH (defined in various studies by the original [26], revised [27], or simplified [28] International Autoimmune Hepatitis Group [IAIHG] criteria) and cholangiographic or histologic features of PSC [29]. AIH-PSC has been identified in 1.4–17% of adults with PSC [29]. Among adults with AIH, the prevalence of AIH-PSC depends upon the presence (44%) or absence (8%) of comorbid inflammatory bowel disease [30]. This syndrome is more common in children, where it has been termed ASC: Up to 50% of children with AIH have cholangiographic features of PSC [31]. In AIH-PSC, as in AIH-PBC, a treatment regimen combining UDCA and corticosteroids (with or without AZA) has been used most frequently, although with a variable clinical response [32,33,34,35,36,37,38]. UDCA alone [33,38,39,40], or corticosteroids with or without AZA [3,33,35,40,41,42], have also been used, albeit less often. Similarly, a combination of UDCA and corticosteroids (with or without AZA) is the most common treatment regimen for ASC [33,36,43,44], while UDCA monotherapy has been used in some cases [33,43,44]. Limited experience using MMF as second-line therapy for patients unresponsive to, or intolerant of, corticosteroids or AZA suggests a greater efficacy for AIH-PSC [16] than for ASC [45]. At 10 years, overall survival is higher for ASC (89%) [36] than for AIH-PSC (~67%) [46], while transplant-free survival is comparable between the two (65% vs. 63%) [36,40].

PBC-PSC [47,48,49,50,51,52,53] and AIH-PBC-PSC [50,54] have also been reported in the literature, although the prevalence of these overlap syndromes is unknown. While there are as yet no standardized diagnostic criteria for these disorders, PBC-PSC is generally characterized by a cholestatic laboratory profile, AMA positivity and/or histologic characteristics of PBC, and cholangiographic and/or histologic features of PSC. AIH-PBC-PSC contains the features above in addition to biochemical, immunologic, and/or histologic evidence of AIH. PBC-PSC has been treated primarily with UDCA monotherapy, although prednisolone/AZA or adalimumab have been used concurrently with UDCA in patients with comorbid rheumatoid arthritis or psoriatic arthritis, respectively [48,49]. While prognostic data on PBC-PSC are sparse, one patient who presented with cirrhosis died from liver failure five years later [50], another patient developed cirrhosis within four years of symptom onset [47], and two patients experienced cholangiographic progression despite biochemical improvement on UDCA monotherapy [48,51]. AIH-PBC-PSC has been treated with either prednisolone and AZA [50], UDCA combined with budesonide and AZA [54], or UDCA combined with prednisone and AZA [55]. The prognosis of this rare disorder is not yet clear.

The rarity of overlap syndromes in autoimmune liver disease diminishes the ease but not the importance of rigorously studying their optimal treatment regimens. Indeed, in many respects overlap syndromes carry worse prognoses than their parent diseases. For example, one cohort study showed that death or liver transplantation occurred in 38% of patients with AIH-PBC but only 19% of patients with PBC during the six-year mean follow-up period (*p* < 0.05) [20]. In a different study, during ~26 months of follow-up, 33% of patients with AIH-PSC experienced liver-related death or transplant, compared to only 8% of those with AIH (*p* = 0.05) [3]. To our knowledge, the primary literature on the treatment of overlap syndromes is devoid of randomized trials and consists entirely of observational cohort studies, case series, and case reports. Zhang et al. published two meta-analyses on the treatment of AIH-PBC [12,56], but both contain serious methodological flaws. Both, for example, mislabel non-randomized cohort studies [4,6,8,10,25,57,58] as randomized controlled trials, and double-count study participants by including a pair of studies with overlapping patient cohorts [4,6]. The more recent of the two meta-analyses [12] purports to examine the effects of UDCA/budesonide combination therapy, but includes several studies in which budesonide is never mentioned [8,10,25].

The treatment recommendations of the American Association for the Study of Liver Diseases (AASLD) and the European Association for the Study of the Liver (EASL) for overlap syndromes in autoimmune liver disease are relatively sparse, reflecting a thin base of primary evidence. For example, the EASL 2017 PBC guidelines state that “Patients with PBC and typical features of AIH *may* (emphasis added) benefit from immunosuppressive treatment in addition to UDCA” and recommend that immunosuppression be given, or considered, in patients with severe or moderate interface hepatitis, respectively [59]. The AASLD 2018 PBC guidelines concede that “the clinical benefit and harm of adding immunosuppressive medications to PBC patients with AIH features require further study”, and recommend tailoring pharmacotherapy to the predominant histologic pattern of injury [60]. For AIH-PSC and ASC, the AASLD 2010 PSC guidelines recommend “corticosteroids and other immunosuppressive agents”, while acknowledging that the impact of these medications remains unclear [29]. The EASL 2015 AIH guidelines further recommend that the addition of UDCA to immunosuppression can be considered, although “It is difficult to draw any firm conclusions because of the small number of patients, the usually retrospective nature of the studies and the heterogeneity of the regimens” [31]. No guidelines exist for the optimal treatment of AIC, PBC-PSC, or AIH-PBC-PSC.

Given the ambiguity of current guidelines, the limited power of individual studies, and dubious quality of existing meta-analyses on the pharmacotherapy of overlap syndromes, an updated synthesis of the primary literature is warranted to guide optimal treatment strategies. We therefore conducted a broad systematic review and meta-analysis of all medications used to treat AIH-PBC, AIH-PSC, PBC-PSC, AIH-PBC-PSC, AIC, or ASC, comparing the efficacy of different treatment regimens for each overlap syndrome as measured by symptomatic, biochemical, histologic, and transplant-free survival outcomes.

## 2. Materials and Methods

This systematic review and meta-analysis are reported in accordance with the Preferred Reporting Items for Systematic Reviews and Meta-Analyses (PRISMA) statement [61].

### 2.1. Selection Criteria

For our systematic review, we included only those studies published as full-text articles in peer-reviewed journals, either in English or with an accessible English translation. Randomized trials and observational cohort studies were accepted, while case reports, case series, and review articles were not. Studies were required to compare two or more pharmacotherapies in human subjects for the first-line treatment of at least one of the following overlap syndromes: AIH-PBC, AIH-PSC, PBC-PSC, AIH-PBC-PSC, AIC, or ASC. Eligible studies also had to report clinical outcomes using at least one of the following parameters: Symptomatic improvement, biochemical improvement, improvement in histologic activity, non-progression of liver fibrosis, or transplant-free survival. There were no restrictions on the age or ethnicity of study participants, the diagnostic criteria used to define overlap syndromes, or the definitions of clinical outcomes.

### 2.2. Data Sources and Search Strategy

Our detailed search strategies were designed with input from an experienced medical librarian and can be found in Table A1. The medical databases Cochrane Library, EMBASE, PubMed, and Web of Science were queried from inception through 30 September, 2019 using the following search terms (and synonyms thereof) in various combinations: “autoimmune liver disease”, “overlap syndrome”, “primary biliary cholangitis”, “primary sclerosing cholangitis”, “autoimmune hepatitis”, “autoimmune cholangitis”, and “autoimmune sclerosing cholangitis”. The titles, abstracts, and/or full texts of the resulting studies were screened by one reviewer (B.F.) to determine their eligibility for the systematic review.

### 2.3. Data Extraction and Quality Assessment

The same reviewer (B.F.) extracted the following data from all included studies: Authorship; year of publication; study design; overlap syndrome(s) examined; diagnostic criteria used; duration of follow-up; treatments administered; number, age, and gender distribution of patients in each treatment group; which clinical outcome(s) were reported (and how they were defined); and the proportion of patients in each treatment group who experienced these outcomes. Whenever feasible, patients who crossed between different treatment regimens during follow-up were censored from the analysis. The quality of individual studies was assessed using the Newcastle–Ottawa Scale [62], which awards a maximum of 9 points (indicating maximal study quality), subdivided by patient selection (0–4 stars), comparability of patient cohorts (0–2 stars), and clinical outcomes and follow-up (0–3 stars). For the latter category of the Newcastle–Ottawa Scale, we considered 2 months, 6 months, and 5 years to be sufficient durations of follow-up for biochemical, histologic, and transplant-free survival outcomes to occur, respectively. Furthermore, we felt that significant attrition bias was unlikely as long as ≤10% of patients were lost to follow-up in a given study.

### 2.4. Statistical Analysis

Data from individual studies were meta-analyzed using a random effects model, and heterogeneity between studies was quantified with the I^2^ statistic (where I^2^ > 50% was considered significant heterogeneity). Separate meta-analyses were performed for each distinct combination of overlap syndrome, treatment comparison, and clinical outcome. Funnel plots were constructed to assess the likelihood of publication bias in all meta-analyses comprising ≥10 studies. All statistical tests were 2-tailed, with a significance threshold of *p* < 0.05. All statistical analyses were performed using Review Manager 5.3 (The Cochrane Collaboration, Copenhagen, Denmark).

### 2.5. Quality of Evidence

We used the Grading of Recommendations Assessment, Development, and Evaluation (GRADE) system to assess the quality of evidence derived from each of our meta-analyses [63]. According to the GRADE system, evidence is rated as either high, moderate, low, or very low quality depending on a number of factors. Meta-analyses of randomized trials are considered high quality by default, but may be downgraded by one or more levels if their constituent trials exhibit poor study design, excessive heterogeneity of results, limited generalizability, low precision, or publication bias. Observational studies are considered by default to be of low quality, and may be downgraded further for the reasons above or upgraded if specific features are present, such as a large magnitude of effect or a dose-response gradient.

## 3. Results

Our systematic literature search yielded 5483 unique publications, of which we included 28 in our systematic review and 20 in our meta-analyses (Figure A1). All publications meeting inclusion criteria for the systematic review were observational cohort studies. The systematic review comprised 20 studies pertaining to AIH-PBC [6,7,8,10,11,13,25,57,64,65,66,67,68,69,70,71,72,73,74,75], two pertaining to AIH-PSC [35,41], two pertaining to AIC [23,24], three pertaining to ASC [33,43,44], and one pertaining to both AIH-PBC and AIC [22]. Of the 21 studies that included patients with AIH-PBC, 17 compared UDCA monotherapy to UDCA + [corticosteroids and/or antimetabolites (AZA or MMF)], i.e., “combination therapy” [6,7,8,10,11,25,57,65,66,67,68,69,70,71,73,74,75]. Five out of these 17 studies also contained a third group of patients treated with corticosteroids ± AZA, thereby allowing additional treatment comparisons [8,68,71,73,74]. Of the four remaining studies of AIH-PBC, one compared UDCA to placebo [13], two compared UDCA to corticosteroids ± AZA [22,64], and one compared corticosteroids + AZA to combination therapy [72]. Both studies of AIH-PSC compared corticosteroids ± AZA to combination therapy [35,41]. Of the three studies pertaining to AIC, two compared UDCA to corticosteroids ± AZA [22,24], while the third compared combination therapy to a complex personalized regimen consisting of UDCA, prednisolone, AZA, MMF, budesonide, rifampicin, and several other agents [23]. All studies of ASC compared UDCA to combination therapy [33,43,44]. Among the 21 AIH-PBC studies in our systematic review, four were ineligible for meta-analysis: One because it was the only study comparing UDCA to placebo [13], one because of an ambiguous overlap between treatment groups [22], and two because no clinical outcomes occurred during follow-up [69,72]. Neither study of AIH-PSC could be meta-analyzed [35,41], because biochemical improvement, the only clinical outcome adjudicated by both studies, occurred in only one of them [35]. One study of AIC was ineligible because it was the only study examining the complex multi-agent regimen referenced above [23], and one study of ASC was ineligible because no clinical outcome was reported for both treatment groups [33].

There were notable studies of each overlap syndrome that did not meet inclusion criteria for our systematic review. The absence of ≥2 distinct treatment groups [2,3,5,18,21,32,34,36,37,38,40,42,76] or the failure to stratify clinical outcomes by treatment group [9,19,20,46,58] were the most common reasons for exclusion. Three studies examining the impact of MMF as second-line therapy for AIH-PBC [16], AIH-PSC [15], AIC [15], and ASC [15,45] were also excluded. We included the larger and more recent [6] of two AIH-PBC studies by Chazouilleres et al. [4,6], given that their patient cohorts overlapped substantially. Furthermore, a retrospective cohort study of six patients with AIH-PSC was excluded for the lack of sufficient detail in its reported clinical outcomes [39]. Lastly, PBC-PSC and AIH-PBC-PSC are not represented in the systematic review because only case series [48,50,55] and case reports [47,49,51,52,53,54] of these syndromes were identified in our literature search.

### 3.1. Characteristics of Included Studies

The characteristics of individual studies in our systematic review are displayed in Table 1. The meta-analysis portion of our review includes 17 studies of AIH-PBC, comprising a total of 402 patients followed at medical centers in France [6,7,8,10,65], China [11,66,68,69,73], Turkey [10,25,57,65], Sweden [10,22,65], South Korea [70], Japan [71,74,75], Tunisia [64,72], Italy [10,65], and the US [10,13,65,67]. These studies ranged in sample size from five to 88 patients, and in mean or median follow-up time from 10 to 119 months. The mean or median age of patients in each study ranged from 38 to 56 years, and 82.5% of patients with AIH-PBC across all studies were female. Twelve out of the 17 studies defined AIH-PBC using the Paris criteria [4]. All 17 studies reported biochemical improvement as a clinical outcome, although in only eight studies did this encompass cholestatic (e.g., AP or GGT) as well as hepatocellular (e.g., alanine aminotransferase [ALT] or aspartate aminotransferase [AST]) markers. Furthermore, six studies provided no a priori definition of biochemical improvement as an endpoint. Four out of the 17 studies reported improvement in histologic activity [6,57,64,75] but because only one of them reported it for ≥2 treatment groups [6], this outcome could not be meta-analyzed. Symptomatic improvement, non-progression of liver fibrosis, and transplant-free survival were reported in three, four, and 10 of the 17 studies, respectively.

Twenty-four patients with AIC were included in the meta-analysis of which four are from a study in Sweden [22] and 20 from a study in the US [24]. The former had a mean follow-up period of 127 months, while the latter did not report duration of follow-up. These patients were a mean of 47.4 years old, and 86% were female. Both studies defined AIC as biochemical and/or serologic evidence of AIH with biochemical and/or histologic evidence of PBC in the absence of anti-mitochondrial antibodies. Biochemical improvement was reported in both studies (though without an a priori definition) and encompassed only hepatocellular, not cholestatic, biomarkers. One of the two studies reported improvement in histologic activity [24], and neither study reported symptomatic improvement, fibrosis non-progression, or transplant-free survival.

Lastly, the meta-analysis includes two studies comprising 25 patients with ASC. One of these studies was conducted in the Czech Republic and followed 11 patients (six female, five male) for a median of 144 months [44]. The other, conducted in Italy, followed 14 patients for a median of 79 months and did not report their gender distribution [43]. The median ages of patients in the former and latter studies were 14 and 9.9 years, respectively. Both studies used a form of the IAIHG criteria to define the AIH component of ASC and for the PSC component, one study required biochemical cholestasis with characteristic findings on either histology or cholangiography [43], while the other study required just cholangiographic findings [44]. Both studies reported biochemical improvement but failed to define it in any detail. Neither study reported symptomatic improvement, improvement in histologic activity, fibrosis non-progression, or transplant-free survival.

### 3.2. Quality of Included Studies

The quality of individual cohort studies in our systematic review was quantified using the Newcastle–Ottawa Scale [62] as shown in Table 2. Lindgren et al. was scored separately for AIH-PBC and AIC, therefore, the effective total number of studies referred to in this paragraph is 29 rather than 28 [22]. Twenty-six out of these 29 studies—20/21 AIH-PBC studies, 2/2 AIH-PSC studies, 1/3 AIC studies, and 3/3 ASC studies—scored 7 out of 9 possible points. None of these 26 studies received the 1–2 points corresponding to comparability of patient cohorts because they did not control for possible confounders in their design or analysis. One cohort study of AIH-PBC nested within a randomized controlled trial did receive an additional point for patient cohort comparability, thereby scoring 8 points [13]. Two studies of AIC scored 6/9 points and in addition to losing 2 points for comparability of patient cohorts, one study lost a point for failing to report the duration of follow-up [23], while the other lost a point for having >10% patients lost to follow-up (treatment outcomes were reported for only four out of eight patients, for unclear reasons) [22].

### 3.3. Clinical Outcomes

#### 3.3.1. AIH-PBC

When comparing combination therapy to UDCA in patients with AIH-PBC, no differences were seen in the rates of symptomatic improvement (RR = 0.75, 95% CI 0.25–2.22, *p* = 0.60, I^2^ = 79%), biochemical improvement (RR = 1.30, 95% CI 0.90–1.87, *p* = 0.16, I^2^ = 57%), non-progression of liver fibrosis (RR = 1.40, 95% CI 0.61–3.21, *p* = 0.42, I^2^ = 77%), or transplant-free survival (RR = 1.06, 95% CI 0.82–1.37, *p* = 0.65, I^2^ = 53%) (Figure 1, Figure 2, Figure 3 and Figure 4). When Fan and Levy et al. were excluded as a sensitivity analysis to simplify the combination therapy arm to UDCA + corticosteroids ± AZA, there remained no difference in biochemical improvement between treatment groups (Figure A2). The same was observed when only studies using the Paris criteria to define AIH-PBC were included in the meta-analysis (Figure A3). When the meta-analysis was restricted to studies where biochemical improvement was defined by both hepatocellular and cholestatic biomarkers, there was a trend toward more biochemical improvement with combination therapy than with UDCA alone (RR = 1.34, 95% CI 0.93–1.93, *p* = 0.12, I^2^ = 27%) (Figure A4). In a sensitivity analysis including only studies with a mean or median follow-up period of >90 months, transplant-free survival was greater in patients receiving combination therapy than in those receiving UDCA alone (RR = 6.50, 95% CI 1.47–28.83, *p* = 0.01, I^2^ = 0%) (Figure A5). When comparing corticosteroids ± AZA to UDCA for the treatment of AIH-PBC, there was no difference in the rate of biochemical improvement (RR = 1.14, 95% CI 0.48–2.72, *p* = 0.76, I^2^ = 0%) or transplant-free survival (RR = 0.96, 95% CI 0.57–1.63, *p* = 0.88, I^2^ = 5%) (Figure 5 and Figure 6). There was a non-significant trend toward superiority of combination therapy over corticosteroids ± AZA with respect to biochemical improvement (RR = 4.00, 95% CI 0.93–17.18, *p* = 0.06, I^2^ = 60%) but not transplant-free survival (RR = 2.03, 95% CI 0.28–14.91, *p* = 0.49, I^2^ = 74%) (Figure 7 and Figure 8).

Of the four studies of AIH-PBC in our systematic review that were not meta-analyzed, three are of limited analytical utility: Lindgren et al. because of overlapping treatment groups [22], and Liu et al. and Serghini et al. because neither study saw biochemical improvement in any of its participants [69,72]. The fourth study noted improved histologic activity in three of nine patients treated with UDCA (13–15 mg/kg/d) versus zero of two patients treated with placebo over a median follow-up of 84 months, but this difference was not statistically significant (Table 3) [13].

#### 3.3.2. AIH-PSC

Although neither study was eligible for meta-analysis, Luth et al. and McNair et al. compared combination therapy to corticosteroids ± AZA for the treatment of AIH-PSC. Luth et al. followed 16 patients for a median of 144 months, and observed biochemical improvement in all six patients treated with combination therapy and nine of 10 patients treated with corticosteroids ± AZA. They did not report other treatment outcomes [35]. McNair et al. followed five patients for a median of 84 months, and observed symptomatic improvement in two of two patients treated with combination therapy and two of three patients treated with prednisolone + AZA. No biochemical improvement was observed in either treatment group. Histologic activity was improved in two of the three patients treated with prednisolone + AZA but unchanged in the one patient treated with combination therapy who underwent a second liver biopsy. No progression in liver fibrosis was seen in this patient, nor in two out of the three patients treated with prednisolone + AZA. Aside from one patient in the prednisolone + AZA group who died from liver failure after six years of follow-up, all patients survived without need for liver transplantation (Table 3) [41].

#### 3.3.3. AIC

When two studies comprised of 20 patients with AIC were meta-analyzed to compare corticosteroids ± AZA to UDCA alone, no difference was seen in the rate of biochemical improvement (RR = 0.59, 95% CI 0.16–2.17, *p* = 0.43, I^2^ = 0%) (Figure 9). Excluded from the meta-analysis given its unique treatment comparison, Campos et al. reported biochemical improvement in one AIC patient treated with UDCA (750 mg/d) and prednisolone, while another patient failed to realize symptomatic or biochemical improvement despite a complex regimen including several rescue therapies (Table 3) [23].

#### 3.3.4. ASC

When two studies comprised of 25 patients with ASC were meta-analyzed to compare combination therapy to UDCA alone, no differences in biochemical improvement (RR = 1.07, 95% CI 0.66–1.74, *p* = 0.78, I^2^ = 0%) or transplant-free survival (RR = 1.00, 95% CI 0.69–1.45, *p* = 1.00, I^2^ = 0%) were identified between the two treatment groups (Figure 10 and Figure 11). Excluded from the meta-analysis because of incomplete reporting of treatment outcomes, Gregorio et al. observed biochemical improvement in 20 of 23 AIC patients treated with combination therapy and improved histologic activity in all three patients treated with UDCA alone (Table 3) [33].

### 3.4. Quality of Evidence

The treatment effect estimates derived from each of our meta-analyses constitute very low quality evidence, given their reliance solely on observational studies and the failure of those individual studies to control for potentially confounding variables (i.e., poor study design by GRADE criteria) [63]. Another common limitation was imprecision due to suboptimal information size, which affected the following meta-analyses: UDCA vs. corticosteroids ± AZA for biochemical improvement in AIC, UDCA vs. combination therapy for biochemical improvement in ASC, UDCA vs. combination therapy for symptomatic improvement in AIH-PBC, UDCA vs. corticosteroids ± AZA for biochemical improvement in AIH-PBC, UDCA vs. combination therapy for fibrosis non-progression in AIH-PBC, UDCA vs. corticosteroids ± AZA for transplant-free survival in AIH-PBC, and corticosteroids ± AZA vs. combination therapy for transplant-free survival in AIH-PBC. Furthermore, the meta-analyses of UDCA vs. combination therapy for biochemical improvement and transplant-free survival in AIH-PBC may have suffered from publication bias (Figure 12) and the studies meta-analyzed in the comparison of UDCA to combination therapy for transplant-free survival in ASC had follow-up times that were too short for deaths or liver transplantations to occur.

## 4. Discussion

Overlap syndromes in autoimmune liver disease are morbid conditions whose rigorous study has been made difficult by their rarity in the general population. Despite numerous small studies over the past few decades, uncertainty remains regarding optimal treatment strategies for these syndromes. Through conducting the above meta-analyses and systematic review of the primary literature, we therefore endeavored to collate all comparative data on first-line therapies for AIH-PBC, AIH-PSC, PBC-PSC, AIH-PBC-PSC, AIC, and ASC, and to determine which therapies, if any, were more effective than others. Ultimately, we demonstrated that combination therapy with UDCA and immunosuppression may be superior both to UDCA alone and to corticosteroids ± AZA for the treatment of AIH-PBC with respect to biochemical improvement and transplant-free survival. The studies of non-AIH-PBC overlap syndromes were either ineligible for our systematic review (PBC-PSC, and AIH-PBC-PSC) or meta-analysis (AIH-PSC) or showed no treatment effect when meta-analyzed (AIC and ASC), highlighting the importance of further primary research in this area.

The most studied treatments for AIH-PBC have been done with UDCA alone, or a combination of UDCA with corticosteroids and/or other immunosuppressive agents (e.g., MMF, AZA). Interest in comparing these two treatment options stems at least in part from a desire to avoid treating patients unnecessarily with immunosuppressive medications that may cause harmful side effects [58]. Our initial meta-analysis of this treatment comparison, pooling 349 patients across 15 studies with AIH-PBC, did not show a difference in biochemical improvement between the two treatment groups.

While we believe this null result should prompt additional studies, it does not preclude the possibility that combination therapy may in fact be superior to UDCA alone for patients with AIH-PBC, or at least a subset thereof. In a cohort of 88 patients with AIH-PBC, Ozaslan et al. showed that the presence of severe interface hepatitis on liver biopsy was an independent predictor of failure to achieve biochemical remission on UDCA monotherapy but not on combination therapy [10]. Hence, adding immunosuppressive therapy to UDCA may be of significant benefit in AIH-PBC with severe interface hepatitis but of little or no benefit in patients with lesser degrees of interface hepatitis, which is a concept highlighted by the EASL 2017 PBC guidelines [59]. Since most studies comparing UDCA to combination therapy in AIH-PBC have been comprised of patients with a wide range of histologic severity and have not stratified treatment outcomes by degree of baseline interface hepatitis, a subgroup analysis to assess this hypothesis further was not feasible in our meta-analysis, and a treatment effect could therefore have been masked.

The fact that our sensitivity analysis including only those studies which defined AIH-PBC using the Paris criteria did not reveal a difference in biochemical improvement between treatment groups suggests that if the null finding of our meta-analysis represented type II error, the error was not driven by excessive heterogeneity in the definition of AIH-PBC. Several studies of AIH-PBC that reported biochemical improvement as a clinical outcome either failed to define it altogether [67,68] or defined it only in terms of hepatocellular markers [6,7,22,25,66,69,70,71,75]. Intuitively, for an overlap syndrome driven by both hepatocellular inflammation and cholestasis, it did not seem as though improved hepatocellular markers alone would make a very reliable surrogate endpoint for longer-term outcomes (e.g., transplant-free survival). To maximize the uniformity and clinical meaningfulness of biochemical improvement as an outcome for AIH-PBC patients, we therefore performed a second sensitivity analysis including only those studies which defined biochemical improvement in terms of both hepatocellular and cholestatic markers [8,10,11,57,65,73,74]. The latter comparison included 202 patients across seven studies, and revealed a non-significant trend toward the superiority of combination therapy over UDCA (RR = 1.34, 95% CI 0.93–1.93). This finding is considered exploratory, since the sensitivity analysis was not pre-specified.

Non-progression of liver fibrosis was reported in relatively few patients with AIH-PBC. Therefore, we were able to meta-analyze only two studies with a total of 46 patients, comparing UDCA to combination therapy [6,10]. With consequently limited statistical power, our meta-analysis did not detect a difference in non-progression of liver fibrosis between treatment groups. However, it is worthwhile noting that one of the two studies, the only one without cross-over of patients between treatment groups, found a significant benefit of combination therapy relative to UDCA (RR = 2.20, 95% CI 1.02–4.74) [6]. The null result of the second study may have been confounded by the cross-over of nine patients from the UDCA group to the combination therapy group, most of whom had no intervening liver biopsy [10]. Therefore, progression of liver fibrosis may have occurred in several of these individuals while on UDCA monotherapy and been subsequently misattributed to combination therapy after they switched between treatment groups.

While we found no difference in transplant-free survival between UDCA and combination therapy in a meta-analysis of 170 AIH-PBC patients across 10 studies, our confidence in this result was limited by the relatively small number of deaths and liver transplants that occurred. Four of the individual studies, for example, tallied ≤1 death or liver transplant each [6,8,57,71]. This phenomenon may have resulted from a combination of having an inadequate duration of follow-up, inadequate study sizes, or disproportionate enrollment of patients early in their disease course with favorable prognoses. When the meta-analysis was restricted to the two studies with a follow-up period >90 months [74,75], combination therapy was associated with a significantly higher transplant-free survival than UDCA alone (RR = 6.50, 95% CI 1.47–28.83). This finding should be regarded as hypothesis-generating, given that the sensitivity analysis above was not pre-specified. Furthermore, because both studies transpired at Japanese medical centers, the generalizability of this result may be hindered by ethnic homogeneity.

Few studies of overlap syndromes reported patients’ symptoms and how they changed with treatment and indeed we were able to meta-analyze only two studies (including 40 patients with AIH-PBC), comparing UDCA to combination therapy with respect to symptomatic improvement [25,66]. Reported symptoms of AIH-PBC included pruritis [25], fatigue [25,66], jaundice [25,66], weight loss [25,66], arthritis/arthralgia [25,66], myalgia [25], lower limb swelling [66], abdominal pain [66], and nausea [66]. The null result of this meta-analysis could suggest an extra-hepatic etiology of some of these symptoms and while combination therapy may enhance biochemical improvement and transplant-free survival compared to UDCA, both liver-directed therapies may be equivalent (and perhaps no better than placebo) in modifying symptoms of AIH-PBC if the latter originated outside the liver. A similar hypothesis was proposed to explain UDCA’s failure to alleviate PBC-related fatigue in a meta-analysis of several randomized trials [84]. Alternatively, it is plausible that combination therapy is more effective than UDCA alone at alleviating a particular symptom of AIH-PBC, but that this effect was obscured by the design of the two meta-analyzed cohort studies, both of which considered a group of several symptoms together rather than individually [25,66].

As UDCA is generally well tolerated (and thus, there is little perceived disadvantage to adding it empirically to an immunosuppressive regimen), few studies have examined immunosuppressive therapy alone for AIH-PBC. While some case series suggest that corticosteroids ± AZA may suffice to induce remission [3,14], all five cohort studies comparing immunosuppression alone to combination therapy [8,68,71,73,74] showed at least a trend toward more biochemical improvement with combination therapy, which approached statistical significance upon meta-analysis (RR = 4.00, 95% CI 0.93–17.18). With only three studies comprising 30 patients who experienced cumulatively only three deaths or liver transplants [8,71,74], our meta-analysis comparing combination therapy to corticosteroids ± AZA with respect to transplant-free survival detected no difference between treatment groups but should be interpreted cautiously in the context of limited statistical power. Similarly, a paucity of studies comparing UDCA to corticosteroids ± AZA in AIH-PBC with respect to biochemical improvement or transplant-free survival precludes a definitive interpretation of our corresponding meta-analyses, both of which failed to demonstrate the superiority of either treatment regimen.

Our decision to censor patients who crossed between treatment groups (in an effort to maximize statistical uniformity) may have contributed to an underestimate of the true impact of combination therapy (compared to UDCA or immunosuppression alone) on clinical outcomes in AIH-PBC. For example, Wu et al. observed biochemical improvement in zero out of three AIH-PBC patients treated with UDCA, zero out of three treated with prednisone, and six out of six treated with UDCA + prednisone as first-line therapies. This apparent superiority of combination therapy is accentuated if one considers that all six patients from this study who failed monotherapy subsequently responded to UDCA + prednisone as second-line therapy [73]. In this or other scenarios where patients switch from mono- to combination therapy after failing the former, confounding factors (e.g., more advanced liver disease), if present, seem more likely to underestimate than to overestimate the effectiveness of combination therapy.

No difference in biochemical improvement was seen between UDCA and corticosteroids ± AZA in a meta-analysis of two studies comprising 20 patients with AIC. However, combination therapy may be a more effective option for these patients, as suggested by the symptomatic and biochemical remission of nine out of 10 patients with “AIH-AIC” treated with UDCA + corticosteroids ± AZA in a case series conducted by Ozaslan et al. [25]. Furthermore, as noted elsewhere, obtaining a deeper pathophysiologic understanding and a more specific diagnostic definition of AIC may facilitate the optimization of treatment strategies [1].

The few cohort studies of ASC meeting inclusion criteria for our systematic review suggest a fairly positive prognosis for these patients, whether they are treated with UDCA or combination therapy [33,43,44]. In the two meta-analyzed studies, 20 of 21 patients on combination therapy and three of four patients on UDCA experienced biochemical improvement, and all 25 patients survived without needing a liver transplant [43,44]. This contrasts with the 65% transplant-free survival reported by Rodrigues et al. in a case series of 28 ASC patients treated with combination therapy however, the prevalence of cirrhosis on initial presentation was considerably higher (77%) in the latter study, compared to ~16% and 0% in Ferrari and Smolka et al., respectively [36,43,44]. To discern the relative efficacy of different treatment regimens for ASC, more studies are required, ideally with larger patient cohorts and/or longer follow-up periods.

It is difficult to draw meaningful conclusions from the two studies of AIH-PSC in our systematic review, which compared corticosteroids ± AZA to combination therapy and were not eligible for meta-analysis. Luth et al. observed biochemical improvement in almost all patients regardless of treatment group, but did not report histologic outcomes or transplant-free survival, and McNair et al. reported multiple clinical outcomes in only five patients [35,41]. We could not compare the effectiveness of different treatment strategies for PBC-PSC or AIH-PBC-PSC, given that our systematic literature search yielded only case series and case reports of these syndromes.

The most important limitation of our systematic review and meta-analysis is its considerable risk of bias, resulting from the lack of treatment randomization or other techniques to minimize confounding in its constituent studies. This highlights the need for multi-center randomized trials and larger observational studies that employ matched patient selection or multivariable modeling to control for confounding factors (e.g., age, comorbidities, and degree of liver fibrosis or histologic activity). Furthermore, our analysis may have forfeited a degree of nuance by not including clinical outcomes of intermediate severity reported in some studies (e.g., decompensated cirrhosis) [11,43]. Lastly, our systematic review examined only first-line pharmacotherapies. Given increasing experimentation with MMF and calcineurin inhibitors as rescue therapies for overlap syndromes [8,10,15,16,17,45], a subsequent review of second-line therapies and their comparative effectiveness may be warranted.

## 5. Conclusions

UDCA, immunosuppression, or a combination of both have been used as first line therapies for overlap syndromes in autoimmune liver disease, but evidence supporting one regimen over the others is sparse. Our systematic review and meta-analysis showed no clear differences in clinical outcomes between these treatment regimens in any of the examined overlap syndromes, although the quality of evidence was very low. While awaiting more definitive studies, providers should continue to rely on professional society guidelines and expert opinion in the treatment of these rare diseases.

## Figures and Tables

**Figure 1 jcm-09-01449-f001:**
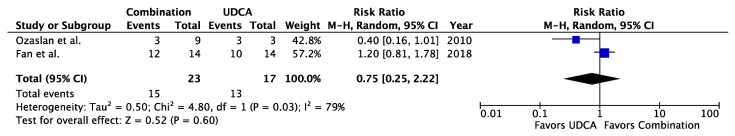
Symptomatic improvement in AIH-PBC patients treated with combination therapy vs. UDCA. UDCA = ursodeoxycholic acid, Combination = UDCA + [corticosteroids and/or antimetabolites].

**Figure 2 jcm-09-01449-f002:**
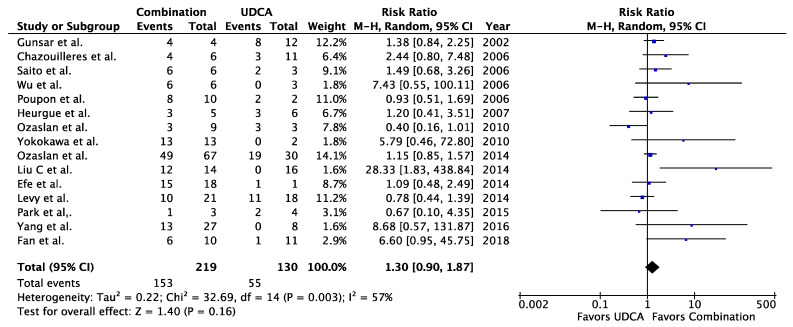
Biochemical improvement in AIH-PBC patients treated with combination therapy vs. UDCA alone. UDCA = ursodeoxycholic acid, Combination = UDCA + [corticosteroids and/or antimetabolites].

**Figure 3 jcm-09-01449-f003:**
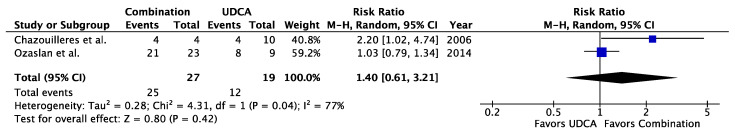
Non-progression of liver fibrosis in AIH-PBC patients treated with combination therapy vs. UDCA alone. UDCA = ursodeoxycholic acid, Combination = UDCA + corticosteroids ± AZA.

**Figure 4 jcm-09-01449-f004:**
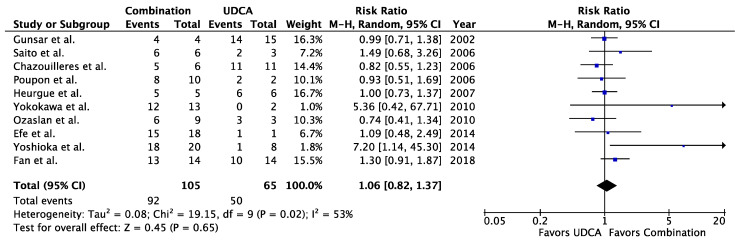
Transplant-free survival in AIH-PBC patients treated with combination therapy vs. UDCA alone. UDCA = ursodeoxycholic acid, Combination = UDCA + [corticosteroids and/or antimetabolites].

**Figure 5 jcm-09-01449-f005:**
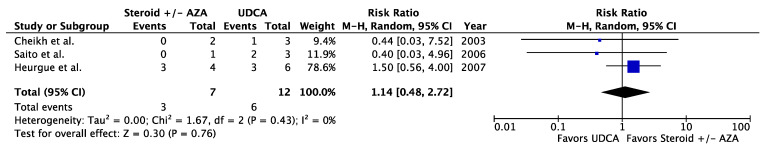
Biochemical improvement in AIH-PBC patients treated with immunosuppression vs. UDCA. UDCA = ursodeoxycholic acid, steroid = corticosteroids, AZA = azathioprine.

**Figure 6 jcm-09-01449-f006:**
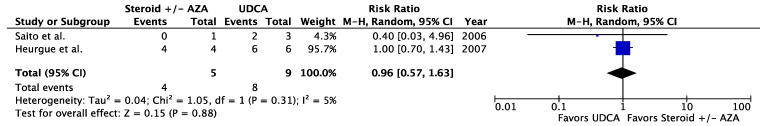
Transplant-free survival in AIH-PBC patients treated with immunosuppression vs. UDCA. UDCA = ursodeoxycholic acid, steroid = corticosteroids, AZA = azathioprine.

**Figure 7 jcm-09-01449-f007:**
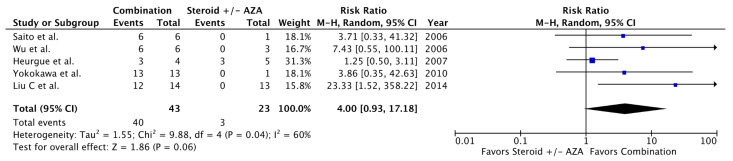
Biochemical improvement in AIH-PBC patients treated with combination therapy vs. immunosuppression. Steroid = corticosteroids, AZA = azathioprine, Combination = UDCA + corticosteroids ± AZA.

**Figure 8 jcm-09-01449-f008:**
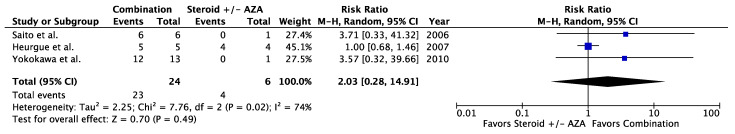
Transplant-free survival in AIH-PBC patients treated with combination therapy vs. immunosuppression. Steroid = corticosteroids, AZA = azathioprine, Combination = UDCA + corticosteroids ± AZA.

**Figure 9 jcm-09-01449-f009:**
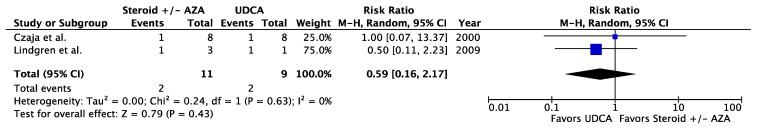
Biochemical improvement in AIC patients treated with immunosuppression vs. UDCA. Steroid = corticosteroids, AZA = azathioprine, and UDCA = ursodeoxycholic acid.

**Figure 10 jcm-09-01449-f010:**
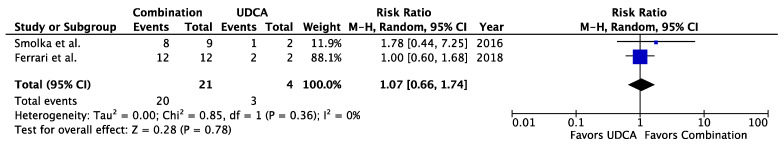
Biochemical improvement in ASC patients treated with combination therapy vs. UDCA. UDCA = ursodeoxycholic acid and Combination = UDCA + prednisone ± AZA.

**Figure 11 jcm-09-01449-f011:**
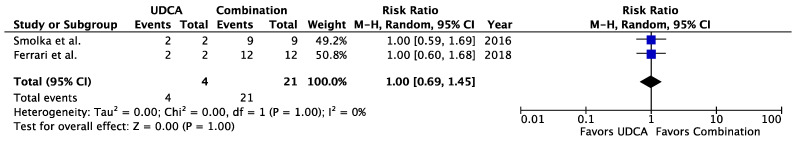
Transplant-free survival in ASC patients treated with combination therapy vs. UDCA. UDCA = ursodeoxycholic acid and Combination = UDCA + prednisone ± AZA.

**Figure 12 jcm-09-01449-f012:**
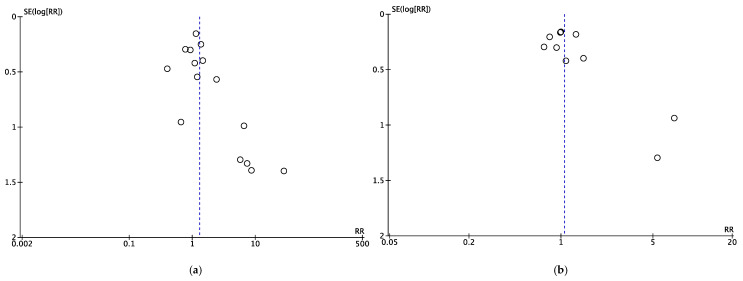
Funnel plots of AIH-PBC studies meta-analyzed to compare (**a**) biochemical improvement and (**b**) transplant-free survival among patients treated with UDCA vs. combination therapy. UDCA = ursodeoxycholic acid and combination therapy = UDCA + [corticosteroids and/or antimetabolites].

**Table 1 jcm-09-01449-t001:** Studies included in systematic review.

Study	Design	Population	*N*	Treatments	Outcomes	Follow-Up (months)
***AIH-PBC***						
Chazouillères 2006 [6]	Retrospective cohort study	AIH-PBC (Paris criteria) [4]. Median age 41; 88% female	17	- UDCA 13–15 mg/kg/d (n = 11) - UDCA 13–15 mg/kg/d + prednisolone 0.5 mg/kg/d ± AZA 50–100 mg/d (n = 6)	Complete biochemical response (ALT < 2× ULN, IgG < 16g/L); improved histologic activity; fibrosis non-progression; TFS	Median 90
Efe 2014 [65]	Retrospective cohort study	AIH-PBC (Paris criteria) [4]. Mean age 50; 89% female	19	- UDCA 12–15 mg/kg/d (n = 1) - UDCA 12–15 mg/kg/d + prednisone 30–60 mg/d ± AZA 50–150 mg/d (n = 18)	Biochemical remission (normalization or >40% reduction in AP at 1 year, normalization of transaminases); TFS	Mean 50
Fan 2018 [66]	Prospective cohort study	AIH-PBC (Paris criteria) [4]. Median age 60 (UDCA group), 48 (combination therapy group); 89% female	28	- UDCA 13–15 mg/kg/d (n = 14) - UDCA 13–15 mg/kg/d + methylprednisolone 12–40 mg/d ± AZA 50–100 mg/d or MMF (dose not reported, n = 14)	Symptomatic improvement; biochemical remission of AIH features (normalization of ALT, AST, and IgG at 1 year); TFS	Median 18
Gunsar 2002 [57]	Retrospective cohort study	AIH-PBC (histologic, serologic, and biochemical features of both diseases). Median age 44 years, 90% female	16 *	- UDCA 13 mg/kg/d (n = 12) - UDCA 13 mg/kg/d + prednisolone 0.5 mg/kg/d (n = 4)	Biochemical improvement ^†^ (Significant decrease in ALT, AST, AP, and globulin levels); improved histologic activity; TFS	Median 28
Heurgue 2007 [8]	Retrospective cohort study	AIH-PBC (Paris criteria) [4]. Median age 44; 87% female	15	- UDCA 11–14.7 mg/kg/d (n = 6) - Corticosteroids 0.5–1 mg/kg/d ± AZA 1.1–2.0 mg/kg/d (n = 5) - UDCA 11–14.7 mg/kg/d + corticosteroids 0.5–1 mg/kg/d ± AZA 1.1–2.0 mg/kg/d (n = 4)	Complete biochemical response (ALT decreased to <2× ULN, AP, and GGT normalized); TFS	Median 60
Joshi 2002 [8]	Retrospective cohort study	AIH-PBC (Paris criteria) [4]. Median age 46; 94% female	16	- UDCA 13–15 mg/kg/d (n = 12) - Placebo (n = 4)	Improved histologic activity (“standardized scoring system for lobular inflammation”, not otherwise specified)	Median 84
Levy 2014 [67]	Retrospective cohort study	AIH-PBC (Paris criteria) [4]. Median age 50–55; 92% female	39	- UDCA 14–15 mg/kg/d (n = 18) - UDCA 14–15 mg/kg/d + AZA or MMF or prednisone (doses not reported, n = 21)	Complete biochemical response ^†^ (“normalization of liver biochemistries”, not otherwise specified)	Median 38
Lindgren 2009 [22]	Retrospective cohort study	AIH-PBC (AIH—IAIHG revised score [27]; PBC—histology, AMA+). Mean age 56; 88% female	25	- UDCA (dose not reported, n = 18) - Corticosteroids ± aza (doses not reported, n = 15) ^‡^	Biochemical remission (normalization of transaminases)	Mean 168
Liu 2014 [69]	Retrospective cohort study	AIH-PBC (AIH—IAIHG simplified score [28]; PBC—Paris criteria) [4]. Median age 53; 86% female	7 ^§^	- UDCA 13–15 mg/kg/d (n = 6) - UDCA 13–15 mg/kg/d + prednisone 10–60 mg/d ± AZA 50 mg/d (n = 1) ^¶^	“Complete response” = histologic improvement or biochemical response (ALT < 2× ULN, IgG < 15.6 g/L)	Range 9–48
Ozaslan 2010 [25]	Retrospective cohort study	AIH-PBC (Paris criteria) [4]. Median age 44; 92% female	12	- UDCA (dose not reported, n = 3) - UDCA + prednisolone or AZA (doses not reported, n = 9)	Symptom resolution; complete biochemical remission (ALT and AST < 2× ULN, Tbili, and gamma globulin normalization); TFS	Median 32
Ozaslan 2014 [25]	Retrospective cohort study	AIH-PBC (Paris criteria) [4]. Median age 48; 84% female	88	- UDCA 13–15 mg/kg/d (n = 30) - UDCA 13–15 mg/kg/d + prednisone 30–60 mg/d ± AZA 50–150 mg/d (n = 67) ^#^	Biochemical remission (normalization or >40% reduction in AP at 1 year, normalization of transaminases); fibrosis non-progression	Mean 66
Park 2015 [10]	Retrospective cohort study	AIH-PBC (Paris criteria) [4]. Median age 49; 100% female	7 **	- UDCA (dose not reported, n = 4) - UDCA + corticosteroids (doses not reported, n = 3)	Biochemical remission (For UDCA + corticosteroid group: Normalization of transaminases, Tbili, IgG. For UDCA group: AP < 3× ULN, AST < 2× ULN, Tbili ≤ 1 mg/dL within 1 year)	Median 70
Poupon 2006 [70]	Retrospective cohort study	AIH-PBC (Paris criteria) [4]. Mean age 46; 100% female	12	- UDCA 12–15 mg/kg/d (n = 2) - UDCA 12–15 mg/kg/d + prednisone 0.5 mg/kg/d ± AZA 1.5 mg/kg/d (n = 10)	Sustained biochemical remission (ALT ≤ 2× ULN, Tbili < 20 mol/L); TFS	Not reported
Saito 2006 [71]	Retrospective cohort study	AIH-PBC (Paris criteria) [4]. Median age 55; 80% female	10	- UDCA 300–600 mg/d (n = 3) - Prednisolone 30 mg/d (n = 1) - UDCA 300–600 mg/d + prednisolone 4–30 mg/d ± AZA (dose not reported, n = 6)	Biochemical response ^†^ (ALT < 2× ULN); TFS	Median 84
Wu 2006 [73]	Retrospective cohort study	AIH-PBC (AIH—IAIHG revised score [27]; PBC—AASLD guidelines) [77]. Mean age 51; gender not reported	12	- UDCA 13–15 mg/kg/d (n = 3) - Prednisone 50 mg/d (n = 3) - UDCA 13–15 mg/kg/d + prednisone 50 mg/d (n = 6)	Complete biochemical remission ^†^ (transaminases < 2× ULN, significant decrease in AP and GGT)	Not reported
Yang 2016 [73]	Retrospective cohort study	AIH-PBC (AIH—Paris criteria [4]; PBC—histologic, serologic, and biochemical features). Mean age 46; 85% female ^††^	35 ^‡‡^	- UDCA 13–15 mg/kg/d (n = 8) - UDCA 13–15 mg/kg/d + prednisolone 15–50 mg/d (n = 27)	Biochemical remission (Paris-I criteria: AP < 3× ULN, AST < 2× ULN, Tbili normalization)	Median 38
Yokokawa 2010 [74]	Retrospective cohort study	AIH-PBC (Paris criteria) [4]. Mean age 56; 88% female	16	- UDCA 300–600 mg/d (n = 2) - Prednisolone 30 mg/d (n = 1) - UDCA 300–600 mg/d + prednisolone 10–40 mg/d ± AZA (dose not reported, n = 13)	Biochemical remission ^†^ (normalization of ALT, AP); TFS	Median 119
Yoshioka 2014 [75]	Retrospective cohort study	AIH-PBC (biochemical, serologic, and histologic features of both diseases) [75]. Median age 55; 93% female	28	- UDCA (dose not reported, n = 8) - UDCA (dose not reported) + corticosteroids 30mg/d (n = 20)	Biochemical remission (normalization of transaminases); improved histologic activity (Ludwig) [78]; improved piecemeal necrosis (undefined); fibrosis non-progression; TFS	Median 94
Liu 2014 [68]	Prospective cohort study	AIH-PBC (biochemical & histologic features of both diseases). Mean age 56; 33% female	43	- UDCA 13–15 mg/kg/d (n = 16) - Prednisone 50 mg/d (n = 13) - UDCA 13–15 mg/kg/d + prednisone 50 mg/d (n = 14)	Complete biochemical remission ^†^ (undefined)	Mean 10
Cheikh 2003 [64]	Retrospective cohort study	AIH-PBC (Paris criteria) [4]. Median age 38; 100% female	5	- UDCA 13–15 mg/kg/d (n = 3) - Prednisone 30 mg/d + AZA 1–2 mg/kg/d (n = 2)	Symptomatic improvement; complete biochemical response (normalization of ALT, AP, GGT, and Tbili); improved histologic activity; fibrosis non-progression	Mean 17
Serghini 2012 [72]	Retrospective cohort study	AIH-PBC (Paris criteria) [4]. Mean age 53; 100% female	5	- UDCA 15 mg/kg/d + corticosteroids 30 mg/d + AZA 50 mg/d (n = 4) - Corticosteroids 30 mg/d + AZA 50 mg/d (n = 1)	Complete biochemical response (ALT < 2× ULN, normalization of AP and GGT)	Median 11
Luth 2009 [35]	Retrospective cohort study	AIH-PSC (AIH—IAHG revised score [27]; PSC—histologic or cholangiographic features). Mean age 34; 19% female	16	- Corticosteroids ± aza (doses not reported, n = 10) - UDCA + corticosteroids ± AZA (doses not reported, n = 6)	Biochemical response ^†^ (Improvement in ALT at 6 months)	Median 144
McNair 1998 [41]	Prospective cohort study	AIH-PSC (AIH—definite by original IAIHG score [26]; PSC—positive cholangiogram). Median age 20; 20% female	5	- Prednisolone 15–80 mg/d + AZA 50–100 mg/d (n = 3) - UDCA 300 mg 1 − 2x/d + prednisolone 20–30 mg/d + AZA 75–150 mg/d (n = 2)	Symptomatic improvement; improved histologic activity; fibrosis non-progression (Batts & Ludwig) [79]	Median 84
***AIC***						
Czaja 2000 [24]	Prospective cohort study	AIC (serologic and biochemical features of AIH; biochemical or histologic features of PBC but AMA). Mean age 46; 85% female	20	- UDCA 13–15 mg/kg/d (n = 8) - Prednisone ± AZA (doses not reported, n = 8)	Biochemical remission (per Czaja 1991) [80]; improved histologic activity (criteria by Ishak et al.) [81]	Not reported
Campos 2017 [23]	Prospective cohort study	AIC (biochemical features of AIH and PBC, histologic features of PBC, AMA-). Mean age 28.5; 100% female	2	- Prednisolone (dose not reported), then UDCA 750 mg/d (n = 1) - UDCA ≤ 1 g/d, cholestyramine ≤ 16 mg/d, rifampicin ≤ 600 mg/d, naltrexone ≤ 50 mg/d, sertraline ≤ 75 mg/d, hydroxyzine ≤ 25 mg QID, amitriptyline ≤ 25 mg/d, phototherapy, molecular adsorbent recirculating system, prednisolone ≤ 30 mg/d, budesonide ≤ 6 mg/d, AZA ≤ 75 mg/d, MMF ≤ 1.5 g/d (n = 1)	Symptomatic improvement; biochemical remission ^†^ (normalization of ALT, AST, AP, and GGT); TFS	Not reported
Lindgren 2009 [22]	Retrospective cohort study	AIC (biochemical & histologic features of PBC, ANA, or ASMA+, AMA-). Mean age 51; 88% female	4 ^§§^	- UDCA (dose not reported, n = 1) - Corticosteroids ± aza (doses not reported, n = 3)	Biochemical remission (normalization of transaminases)	Mean 127
Smolka 2016 [44]	Retrospective cohort study	ASC (probable or definite AIH by simplified IAIHG score modified for children [82]; positive cholangiogram). Median age 14; 55% female	11	- UDCA 15–20 mg/kg/d (n = 2) - UDCA 15–20 mg/kg/d + prednisone 1–2 mg/kg/d + AZA 1–2 mg/kg/d (n = 9)	Biochemical remission ^†^ (undefined); TFS	Median 144
Ferrari 2018 [43]	Retrospective cohort study	ASC (biochemical and histologic and/or cholangiographic features of PSC; AIH features on IAIHG revised score) [27]. Mean age 9.9; gender not reported	14 ^¶¶^	- UDCA 15–20 mg/kg/d (n = 2) - UDCA 15–20 mg/kg/d + prednisone 1 mg/kg/d + AZA 1.5–2 mg/kg/d (n = 12)	Biochemical remission ^†^ (undefined)	Median 79
Gregorio 2001 [33]	Prospective cohort study	ASC (probable or definite AIH by IAIHG revised score [27]; positive cholangiogram. Median age 11.8; 56% female	26 ^##^	- UDCA (dose not reported, n = 3) - Prednisolone 2 mg/kg/d ± UDCA (dose not reported) ± AZA 1–2 mg/kg/d (n = 23)	Biochemical remission ^†^ (normalization of liver function tests); improved histologic activity (inflammatory activity index scored 0–12, Incorporating portal tract inflammation, lobular activity, and piecemeal necrosis) [33]	Median 72

Transplant-free survival is expressed as the raw fraction of study participants who did not experience death or liver transplant by the end of follow-up. Fibrosis and histologic activity were assessed using the METAVIR scoring system (A0 = absent, A1 = mild, A2 = moderate, and A3 = severe histologic activity; F0 = no fibrosis, F1 = portal fibrosis without septa, F2 = few septa, F3 = numerous septa without cirrhosis, and F4 = cirrhosis) [83] except where otherwise noted. In treatment regimens comprising “corticosteroids”, the authors did not specify which corticosteroid(s) were used. * Excluded 1 patient lost to follow-up and 3 patients who crossed between treatment groups. ^†^ Biochemical endpoint not defined a priori in Methods, though authors may comment in Results on how laboratory values changed with treatment (see parentheses). ^‡^ Authors double-count ≥8 AIH-PBC patients, who actually received both UDCA and corticosteroids, in the UDCA and corticosteroid groups. ^§^ Of the 10 AIH-PBC patients studied, treatment outcomes were reported for only 7. ^¶^ Of the 5 patients who eventually received combination therapy, the 4 who received UDCA monotherapy beforehand were censored. ^#^ Unable to exclude 9 patients from combination therapy group who crossed over from UDCA group, because authors do not distinguish their treatment outcomes from those of non-crossover patients. ** Excluded 2 patients who crossed between treatment groups. ^††^ This percentage pertains to the total of 46 AIH-PBC patients (see below). ^‡‡^ Of the 46 AIH-PBC patients studied, treatment outcomes were reported for only 35. ^§§^ Of the 8 AIC patients studied, treatment outcomes were reported for only 4. Note that 7 (88%) of the total 8 patients were female. ^¶¶^ Excluded 5 patients who crossed between treatment groups. ^##^ Excluded 1 patient with mild disease who did not receive pharmacotherapy. Note that 15 (56%) of the total 27 patients were female. AIH: Autoimmune hepatitis, PBC: Primary biliary cholangitis, PSC: Primary sclerosing cholangitis, AIC: Autoimmune cholangitis, ASC: Autoimmune sclerosing cholangitis, AMA: antimitochondrial antibody, ANA: Antinuclear antibody, ASMA: Anti-smooth muscle antibody, IAIHG: International Autoimmune Hepatitis Group, AASLD: American Association for the Study of Liver Diseases, UDCA: Ursodeoxycholic acid, MMF: Mycophenolate mofetil, ALT: Alanine aminotransferase, IgG: Immunoglobulin G, Tbili: Total bilirubin, ULN: Upper limit of normal, TFS: Transplant-free survival, AST: Aspartate aminotransferase, AP: Alkaline phosphatase, GGT: Gamma-glutamyl transferase.

**Table 2 jcm-09-01449-t002:** Quality of studies included in systematic review as quantified by Newcastle–Ottawa Scale.

Study	Selection				Comparability	Exposure			
	Represent-Ativeness of Exposed Cohort	Selection of Non-Exposed Cohort	Ascertainment of Exposure	Demonstration that Outcomes of Interest Were not Present at Start of Study	Comparability of Cohorts on Basis of Design or Analysis *	Assessment of Outcomes	Length of Follow-Up ^†^	Adequacy of Follow-Up ^‡^	Total
Campos 2017 [23]	*	*	*	*		*		*	6
Chazouillères 2006 [6]	*	*	*	*		*	*	*	7
Cheikh 2003 [64]	*	*	*	*		*	*	*	7
Czaja 2000 [24]	*	*	*	*		*	*	*	7
Efe 2014 [65]	*	*	*	*		*	*	*	7
Fan 2018 [66]	*	*	*	*		*	*	*	7
Ferrari 2018 [43]	*	*	*	*		*	*	*	7
Gregorio 2001 [33]	*	*	*	*		*	*	*	7
Gunsar 2002 [57]	*	*	*	*		*	*	*	7
Heurgue 2007 [8]	*	*	*	*		*	*	*	7
Joshi 2002 [8]	*	*	*	*	*	*	*	*	8
Levy 2014 [67]	*	*	*	*		*	*	*	7
Lindgren 2009 [22]									
AIH-PBC	*	*	*	*		*	*	*	7
AIC	*	*	*	*		*	*		6
Liu 2014 [69]	*	*	*	*		*	*	*	7
Liu 2014 [68]	*	*	*	*		*	*	*	7
Luth 2009 [35]	*	*	*	*		*	*	*	7
McNair 1998 [41]	*	*	*	*		*	*	*	7
Ozaslan 2010 [25]	*	*	*	*		*	*	*	7
Ozaslan 2014 [25]	*	*	*	*		*	*	*	7
Park 2015 [10]	*	*	*	*		*	*	*	7
Poupon 2006 [70]	*	*	*	*		*	* ^§^	*	7
Saito 2006 [71]	*	*	*	*		*	*	*	7
Serghini 2012 [72]	*	*	*	*		*	*	*	7
Smolka 2016 [44]	*	*	*	*		*	*	*	7
Wu 2006 [73]	*	*	*	*		*	* ^§^	*	7
Yang 2016 [73]	*	*	*	*		*	*	*	7
Yokokawa 2010 [74]	*	*	*	*		*	*	*	7
Yoshioka 2014 [75]	*	*	*	*		*	*	*	7

* One point awarded for each variable that was statistically controlled for, up to a maximum of 2 points. The only point awarded was to Joshi et al., given the latter’s study design (retrospective cohort study nested within a randomized controlled trial). ^†^ Criterion to gauge if follow-up time was sufficient for outcomes of interest to occur. For this analysis, 2 months, 6 months, and 5 years were considered sufficient for biochemical, histologic, or transplant-free survival outcomes, respectively. One point was awarded if the study’s follow-up period was adequate for ≥1 of these outcomes and if that outcome was reported. ^‡^ One point awarded if number of subjects lost to follow-up was small enough (≤10% in this analysis) to make attrition bias unlikely. ^§^ Median follow-up not specified, but several patients were followed for ≥2 months and biochemical remission did occur.

**Table 3 jcm-09-01449-t003:** Results of studies included in systematic review but excluded from meta-analysis.

Study	Treatments Compared	Symptom Improvement	Biochemical Improvement	Improved Histologic Activity	Fibrosis Non-Progression	Transplant-Free Survival	Reason for Exclusion from Meta-Analysis
***AIH-PBC***							
Joshi 2002 [8]	UDCA	—	—	3/9	—	— *	Only study comparing UDCA to placebo
	Placebo	—	—	0/2	—	— *	
Lindgren 2009 [22]	UDCA	—	3/18	—	—	—	Overlapping treatment groups (see Table 1)
	Corticosteroids	—	5/15	—	—	—	
Liu 2014 [69]	UDCA	—	0/6	—	—	—	No endpoints reached in either treatment group
	UDCA + prednisone±	—	0/1	—	—	—	
Serghini 2012 [72]	UDCA + corticosteroids + AZA	—	0/4	—	—	—	No endpoints reached in either treatment group
	Corticosteroids + AZA	—	0/1	—	—	—	
***AIH-PSC***							
Luth 2009 [35]	Corticosteroids±	—	9/10	—	—	—	No comparator study for biochemical improvement (see below)
	UDCA + corticosteroids ± AZA	—	6/6	—	—	—	
McNair 1998 [41]	Prednisolone + AZA	2/3	0/3	2/3	2/3	2/3	No biochemical endpoints reached in either group
	UDCA + prednisolone + AZA	2/2	0/2	0/1 ^†^	1/1 ^†^	2/2	
***AIC***							
Campos 2017 [23]	Prednisolone, then UDCA, cholestyramine, rifampicin, naltrexone, sertraline, hydroxyzine, amitriptyline, phototherapy, molecular adsorbent recirculating system, prednisolone, budesonide, AZA, MMF	—	1/1	—	—	1/1	No comparator studies with similar treatment groups
		0/1	0/1	—	—	1/1	
***ASC***							
Gregorio 2001 [33]	UDCA	—	—	3/3	—	—	No endpoint is reported for both treatment groups
	Prednisolone ± UDCA ± AZA	—	20/23	—	—	—	

Transplant-free survival is expressed as the raw fraction of study participants who did not experience death or liver transplant by the end of follow-up. Fibrosis and histologic activity were assessed using the METAVIR scoring system [83] except where otherwise noted. In treatment regimens comprising “corticosteroids”, the authors did not specify which corticosteroid(s) were used. The treatment comparison in Joshi et al. was reported as statistically non-significant; statistical testing for the other studies was not reported. * Transplant-free survival was omitted from this analysis given cross-over of four patients from placebo to the UDCA group after ~2 years. ^†^ Liver biopsy results were reported for only one of the two patients in this treatment group. AIH: Autoimmune hepatitis, PBC: Primary biliary cholangitis, PSC: Primary sclerosing cholangitis, AIC: Autoimmune cholangitis, ASC: Autoimmune sclerosing cholangitis, UDCA: Ursodeoxycholic acid, MMF: Mycophenolate mofetil.

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
