# Peer review of "Treatment of Overlap Syndromes in Autoimmune Liver Disease: A Systematic Review and Meta-Analysis"

_jcm, 2020, doi:10.3390/jcm9051449_

Round 1
Reviewer 1 Report
The authors approach the lack of international accepted guidance for the treatment of hepatic overlap syndromes.
The start their manuscript by appropriately describing all the overlap syndromes and then they lay down the problem of the lack of randomized controlled trials analyzing the treatment of these diseases. The extensive literature research rendered interesting results. There are no large studies approaching the problem and in a very elegant way they demonstrated the lack of agreement among societies and groups devoted to the treatment of these rare conditions.
The authors did an elegant design of literature search and also demonstrated in this manuscript that the quality of papers dealing with these diseases is low and that precisely pinpoints the need for large, multi-center controlled trials. In my opinion the authors should not be shy on stating this on their conclusions.
The text is extensive and sometimes difficult to follow but the tables and graphics are very helpful and self explanatory.
My recommendation is to accept the manuscript for publication.
Author Response
Dear Reviewer,
Thank you very much for your time and consideration in reviewing our manuscript. In accordance with your recommendation to emphasize the importance of conducting randomized controlled trials of the treatment of overlap syndromes in the future, we modified a sentence in the final paragraph of our Discussion section (lines 559-562): we replaced
"Since conducting randomized controlled trials of diseases as rare as the overlap syndromes in autoimmune liver disease may be an unrealistic goal, it will be particularly useful for future observational studies of overlap syndromes to control for confounding variables—such as age, comorbidities, and degree of liver fibrosis or histologic activity—via multivariable modeling or matched patient selection."
with
"This highlights the need for multi-center randomized trials and larger observational studies that employ matched patient selection or multivariable modeling to control for confounding factors (e.g., age, comorbidities, and degree of liver fibrosis or histologic activity)."
Sincerely,
Benjamin L. Freedman, MD
Christopher J. Danford, MD
Vilas Patwardhan, MD
Alan Bonder, MD
Reviewer 2 Report
This is a really good, very comprehensive, worth reading and intelligible manuscript. I cannot point out the strength of this manuscript: it is a meta-analyses/review. So there are no new things in it, but it is really good and comprehensive as stated in my review.
Author Response
Dear Reviewer,
Thank you very much for your time and consideration in reviewing our manuscript.
Sincerely,
Benjamin L. Freedman, MD
Christopher J. Danford, MD
Vilas Patwardhan, MD